# Shopping When You Are Deafblind: A Pre-Technology Test of New Methods for Face-to-Face Communication—Deafblindness and Face-to-Face Communication

**Claude Vincent** [1,2,*], **Walter Wittich** [3,4], **François Bergeron** [1,2], **Mathieu Hotton** [1,2] **and Bertrand Achou** [2,5]

[1] Rehabilitation Department, Université Laval, Quebec City, QC G1V 0A6, Canada; Francois.Bergeron@rea.ulaval.ca (F.B.); Mathieu.Hotton@fmed.ulaval.ca (M.H.)

[2] Centre of Interdisciplinary Research in Rehabilitation and Social Integration (Cirris), Université Laval, Quebec City, QC G1V 0A6, Canada; bertrand.achou@gmail.com

[3] School of Optometry, Université de Montréal, Montreal, QC H3T 1J4, Canada; walter.wittich@umontreal.ca

[4] Centre for Interdisciplinary Research in Rehabilitation of Greater Montreal, Montreal, QC H3S 1M9, Canada

[5] Retirement and Savings Institute, HEC Montréal, Montreal, QC H3T 2A7, Canada

[*] Correspondence: claude.vincent@rea.ulaval.ca

**Abstract:** This article presents the first-year results of a project that aimed to explore the feasibility of using a braille display and a smartphone in society to improve face-to-face communication for a person living with deafblindness, using a simulated communication situation. An applied experimental development design was implemented, followed by a pre-test in the community. Two clinicians and an engineer conducted communication tests with three communication partners with normal vision in a shopping mall. A blind clinician acting as deafblind bought an iPhone case and asked for the location of two stores. Communication partners did not report any difficulties, understood the exchanges, and were proud to have helped a person living with deafblindness. No communication breakdowns or keyboard input incidents occurred. Speech turns were not optimal but can be improved. Clinicians proposed a sequence of three training modules: (1) prior knowledge (basic operations for iPhone, software, and braille display), (2) methods for preparing a face-to-face discussion, and (3) processes during a face-to-face discussion. Results demonstrate the feasibility of using a tactile technological solution coupled with a smartphone to interact with unknown interlocutors. Technology trials form the groundwork for a 9-month case study, involving two individuals with deafblindness.

**Keywords:** deafblindness; dual sensory loss; Usher syndrome; assistive communication technology; iPhone; braille notetaker; living laboratory

## 1. Introduction

Communication is a basic human right, and this article helps to highlight support for communication in society for people who become deafblind in their early fifties. There is a great need for technological solutions for individuals who are profoundly deaf from birth and communicate in sign language when they lose their sight as they get older. These individuals are no longer able to use their usual technology aids that are mainly based on visual communication. Faced with this problem, they often find themselves socially isolated, even from their own family members. It is therefore essential to combine clinical efforts with research to generate solutions that can reduce barriers to social participation [1]. This article presents the first-year results of a research project that aimed to measure effectiveness, impact on social participation, and cost of communication technology alternatives proposed to two people newly living with deafblindness, focusing on the selection of a communication technology and the pre-technology testing in a common communication setting.

In 2018, the World Federation of the Deafblind estimated that 0.2–2% of the world's population were living with combined vision and hearing impairments [2]. Among this population, people with Usher syndrome type 1 (USH1) experience profound hearing loss or deafness at birth and decreased night vision by age 10, progressing to severe vision loss by midlife, as a result of the effects of retinitis pigmentosa [3,4]. They also experience balance problems from birth [5]. Usher syndrome affects approximately three to ten in 100,000 people worldwide [6] or around 4 to 17 in 100,000 people [4].

In 2018, a systematic review focusing on communication technologies enabling internet access for individuals with deafblindness revealed that no scientific study has specifically evaluated their effectiveness [7]. A scoping review published in 2021 [8] examined device abandonment in deafblindness, functioning, and usability through the International Classification of Functioning, Disability and Health model. Results evidence the lack of usability assessments (i.e., measuring satisfaction, effectiveness, and efficiency). From the 10 studies consulted in that recent scoping review [8], usability was challenged in devices that rely on the "other" sense (i.e., you need vision or hearing to use the device). The personal, device-related, environmental, and intervention-based barriers and facilitators to device use have been described for vision impairment [9–12] or hearing impairment [13] separately. Haptic and tactile aids are rarely studied [8], whereby only one study on deafblindness so far emphasised tactile aids [14].

In 2019, Cantin and colleagues [14] conducted a study adopting the "living lab" approach to report on the experience of a person with deafblindness using a communication technology alternative (a braille display with a smartphone) in an everyday community activity, such as going to a restaurant. It is the only research that evaluated the participant and her sighted and hearing communication partners' perceptions of their efficacy in communication interactions in real-life situations with and without a communication technology. The technology used in this study elicited mostly positive attitudes and perceptions by both the participant and her communication partners. There were some recurring technical problems that were critical to effective communication; the technology generated relatively poor communication compared to sign language, but the participant's emotional experience was positive and sustained. Due to her mastery of tactile sign language and braille, the person living with deafblindness was able to use the communication technology alternative with a waitress in her favorite restaurant. The discussions of the results between the members of the study team and the counsellor led to four suggestions for improvements. These were related to (1) invitation and instructions, (2) notification of intent, (3) speech turns, and (4) feedback to the person with deafblindness. The detailed research report captured the technical and communications problems, as well as their recommendations [15].

Based on this very limited scientific literature, the research objectives for the first-year project were: (1) to demonstrate the feasibility of using a braille display and a smartphone in simulated communicational and social exchange; and (2) to propose clinical recommendations for eventually training persons with USH1 for face-to-face communication in society. The setting described in this study is a pre-requisite to further tests of usability by deafblind people. Before measuring usability, we must test the technology in simulating face-to-face communication in a shopping mall, in a real context, with an unfamiliar person.

## 2. Materials and Methods

### 2.1. Applied Research and Experimental Development Design (1st Design)

"Research and experimental development" design is defined as a "creative and systematic work undertaken in order to increase the stock of knowledge—including knowledge of humankind, culture and society—and to devise new applications of available knowledge" [16]. In the first year of the project, the two creative and systematic steps were Selection of a communication technology and Adaptation.

*Selection of a communication technology*: In the first four months, we conducted an internet review of all available communication technologies for deafblindness [17]. This

review covered 19 notetakers/braille displays, 17 apps, 3 phones, and 8 types of software. The analysis of these technologies was conducted through a videoconference focus group of 15 experts (5 researchers, 8 clinicians, 2 research project planning officers, 1 observer from a health funding agency). Before this focus group, experts had received the review [17] including red, orange, and green proposed selection codes for each product (do not retain, perhaps, retain). The facilitator of the focus group followed the document rigorously to have the experts discuss each of the products. At the end, 4 notetaker devices were retained, few apps, one phone, and 3 types of software. Validation of availability evidenced that the choice of suitable equipment was limited; the final technology solution is presented below.

*Adaptation*: It took another six months to complete the adaptation, programming, and ergonomics of the equipment to make sure it would be usable by a French-Canadian person with USH1. That was realised with the support of an engineer research project planning officer, a low-vision therapist who is blind and already knows braille, and a clinician competent in sign language. This was necessary to learn how to teach, train, give explanations, and communicate face-to-face as well as remotely. The engineer made sure that the retained equipment interfaced smoothly so that the person living with deafblindness would know they have turned on their phone (by vibration), that they recognize the communication application selected among the others, that they can check the Bluetooth connection and if the braille notetaker is on, that they can know who is writing to them, and that they can know how to respond. Parameters setting of the selected equipment must be carried out so that the person with deafblindness can quickly issue a communication or receive a written message. A detailed report was written following this applied R&D phase, including the management of the technical solutions [18] and abandonment of a communication solution with fake chat [19].

Technology Solution

The team created an invitation sign for dialogue as shown in Figure 1. Effective to attract the attention of the public, it was printed on laminated paper (5 1/2 inches × 7 1/2 inches) with the text (translated from French): "To help me, TOUCH MY ARM. I am DEAF <u>and</u> BLIND". It was suspended on a necklace worn by the person with deafblindness.

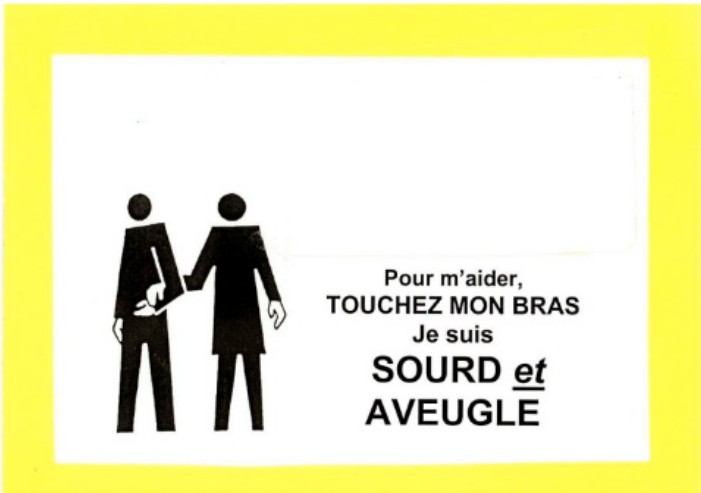

**Figure 1.** Sign inviting dialogue, in French, from [20] (p. 1).

As shown in Figure 2, the technology environment integrates two main devices, a braille notetaker (used by the person with deafblindness) connected to Bluetooth and a smartphone (used by a sighted communication partner). Configurations initially concerned the settings of software systems/applications running on an iPhone X, in particular: the iOS operating system, the VoiceOver screen reader, and the note-editing Notes app. This app is available only on the App Store for iPhone and iPad. This configuration was supplemented

by creating alternate texts associated with keyboard shortcuts running on the iOS operating system.

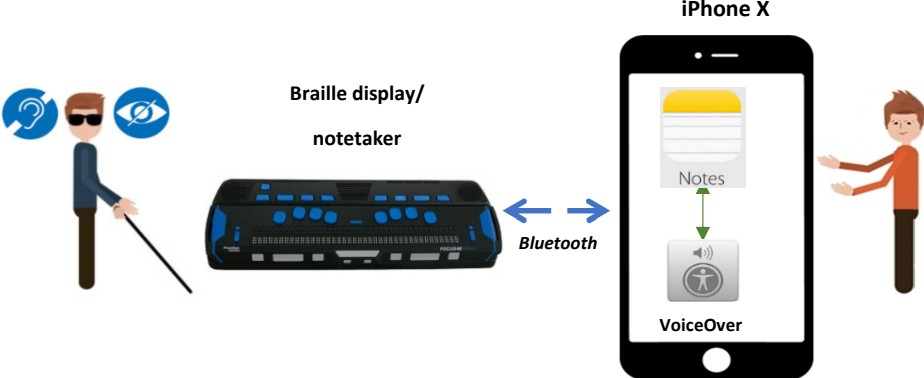

**Figure 2.** Solution for face-to-face communications of people with deafblindness, from [18] (p. 1).

Braille display used was the Focus 40 Blue 5th gen (14.5 × 3 × 0.75 po, 1.24 lbs, $3295 USD). Copyright © Freedom Scientific Inc. Clearwater, United States and Barendrecht, The Netherlands. It is a black braille keyboard with the 8 dots in color blue, 40 cells, and good Bluetooth connection reliability. It was chosen for its adequate number of cells, robustness, USB type C connection, and Bluetooth ability. An iPhone X (iOS 13+) was selected for the smartphone. VoiceOver, the screen reader on iOS, is free. Notes, the face-to-face communication solution, is free because it is included in Apple iOS. Notes was chosen for its accessibility (e.g., turns of speech) and its ergonomics as it works like a word processor. SMS (Short Message Service) apps, such as Messenger or fake chat apps [19], were not retained because it was not possible to actualize turns of speech in face-to-face communication. Mail, the solution for remote communications, was selected for its accessibility through VoiceOver.

### 2.2. Technology Pretest in the Community (2nd Design)

In the last two months, the technology pre-test took place in a shopping mall near the rehabilitation center, followed by a detailed written report including the recommendations for training [20]. In the shopping mall *Place Versailles* (in Montreal, Quebec, Canada), three trials for face-to-face communication were conducted. Test 1—at a smartphone accessory sales kiosk, the person with deafblindness addresses the attendant to ask for an iPhone case; Test 2—in the corridor, the person with deafblindness addresses a person from the public to ask for the location of a one-price store (Dollarama); Test 3—in an information booth, the person with deafblindness addresses the security guard to ask them for the location of a particular female clothing store (Le Garage). Exchanges with Notes were saved on the iOS device. Sighted communication partners were questioned after each test about (1) understanding the invitation message, (2) the occurrence of difficulties, (3) the level of satisfaction with the exchange, and (4) their feelings after exchange.

### 2.3. Participants

There were three R&D participants, employed by two rehab centers located in the Montreal area. The low-vision therapist, who is also blind, simulated the person living with deafblindness in the shopping mall. The special educator, competent in sign language, participated in the test planning and on-site analysis and discussions in the mall. The engineer took notes on the events occurring during and after the tests. The community participants were an attendant at a kiosk, a passer-by, and a security agent. They were the three communication partners (interlocutors) with normal vision that were interviewed after the tests in the shopping mall. Given the type of research design (applied research and experimental development), the same participants that adapted the technology solution in clinics volunteered for the pre-test. This means that "selection bias" is needed here since it

is not a question of having a representative sample but of establishing a proof-of-concept in social context. Since there were no deafblind people in the development phase, a clinician with no vision simulated one. The three partners with normal vision in the shopping mall were chosen spontaneously on the site, upon their availability.

This project was approved by the research ethics board of the Centre for Interdisciplinary Research in Rehabilitation of Greater Montreal (CRIR-1399-0319 multi).

*2.4. Data Analysis*

The text message that appeared on the iOS Notes app was transferred into a grid and translated from French to English with all the mistakes (see Table 1). This was necessary for analysis of the exchanges between the communication partners and the simulated person with deafblindness. The observational field notes taken by the engineer were also analyzed by the R&D team.

**Table 1.** Texts of the messages exchanged between the member of the team playing the role of a person with deafblindness and the communication partners with normal vision during the interaction tests carried out in place Versailles in Montreal (translated from French [20])[1].

| | **Test 1—Attendant at a Kiosk Selling Smartphone Accessories** | **Test 2—Passer-by in the Corridor in front of the Dollarama** | **Test 3—Security Agent in an Information Kiosk** |
|---|---|---|---|
| 1 | Hello, I am deaf and blind. Use this device to communicate with me. Write on the keyboard, but avoid touching long, swiping your finger and do not use accents. At the end of your messages, double-tap on return. Please, make short sentences. Write "ok" when a- you are ready.↵ | Hello (invitation message) Write "ok" when a- you are ready.↵ | Hello (invitation message) Write "ok" when a- you are ready.↵ |
| 2 | Ok | Hello how are y ↵ | Ok |
| 3 | ↵ | ou? ↵ | ↵ |
| 4 | I would like a case for my iphone please↵ | Fine thanks! can you bring me to the entrance of the Dollarama please? ↵ | is Le Garage store here?↵ |
| 5 | ↵ | | Yes↵ |
| 6 | For which iphone↵ | | ↵ |
| 7 | the one in your hands↵ | | which direction please? ↵ |
| 8 | Ok↵ | | Ee f behindinfrontyes↵ |
| 9 | do you have one with a hole or ring to hang a rope?↵ | | ok thank you very much↵ |
| 10 | I will check↵ | | |
| 11 | I have three models what color the 2? ↵ | | |
| 12 | Blue and red | | |
| 13 | is rubber too?↵ | | |
| 14 | plastic↵ | | |
| 15 | ok, the blue please↵ | | |
| 16 | Ok madam↵ | | |
| 17 | what price? It is a 25 but I make you a price 20↵ | | |
| 18 | Ok thank you↵ | | |
| 19 | It's a pleasure↵ | | |

Note: [1] The texts of the person with deafblindness appear in blue and those of the sighted person are in red. Activating the Return button corresponds to the symbol " ↵ " in blue and red and defines row breaks in the table.

## 3. Results

### *3.1. Feasibility*

3.1.1. Quality of Communicational Exchanges on iOS Notes

Analysis of the results in Table 1 showed that spelling and punctuation errors, as well as non-compliance with the instruction to press the "Return" key twice after individual messages, did not have an impact on communications. This instruction was intended to organize messages that make a distinction between the texts of the two communication partners aimed at rereading the discussions. Indeed, as shown in Figure 4a, on the interface of iOS Notes, the texts all have the same color, the paragraph break marks did not appear, and the messages were not separated and arranged in cells of a grid. Tracing the texts of the discussions (turns of speech) *a posteriori* on the interface of iOS Notes was a much more difficult task.

3.1.2. Understanding the Invitation Message and Occurrence of Difficult Times in Communication

Table 1 shows that communication partners had received and understood the invitation message. There were no communication breakdowns or keyboard input incidents (long touching and finger swipes) during the three tests. We noticed that the participants did not always consider the instruction to touch the Return button twice at the end of each message (Test 1—lines 6 to 19, Test 2—line 3, Test 3—lines 4 and 7). In Test 1, two exchanges took place without the "Return" button being touched (lines 11 and 17). The question mark was missing (lines 6 and 7). In Test 3, the answer was contradictory and without space between the words (line 8). Since the respondent thanked the security agent (line 9), maybe he guided the person tactilely by answering *behind* (the person is turning) *in front*.

3.1.3. Level of Satisfaction with the Exchanges and Feelings after Exchanges

After the interaction, the communication partners with normal vision did not report any difficulties, had understood everything about the exchanges, and above all, were proud to have helped a person with deafblindness. In this regard, the person simulating deafblindness let the sighted participants know that this was indeed a simulation and that she could hear and speak.

### *3.2. Clinical Recommendations for Eventually Training*

A technology solution for people with deafblindness is only possible if a training program is actualized for new users of tactile technologies. The clinicians had to become familiar with how technologies work and how to ensure their interconnectivity. They also suggested that it is essential to ensure that a new user has a minimum of braille as well as French grammar, wishes to communicate using an iPhone, and can afford the internet costs. The training in the clinic with the equipment may take between 9 and 12 sessions to complete (once or twice per week, 1 to 2 h per session). After the experimentation, clinicians using tactile sign language proposed a sequence of three training modules [20], which are presented in Table 2.

**Table 2.** Sequence of three training modules proposed to new clients with deafblindness.

Module 1—Prior Knowledge:
- Basic operation of iPhone X, VoiceOver, Notes, and Mail.
- Basic operation of the notetaker/braille display Focus 40.

Module 2—Methods for preparing a face-to-face discussion (with Notes)
1. Unlock the iPhone X.
2. Open the "Notes" application.
3. Press the "Add" button.
4. On the first line, give a meaningful title to the note, then press Return.
5. Write "; b" and space for the introduction to be written (text replacement).
6. Close wallet case on iPhone X. Keep case closed until the start of a face-to-face conversation.

Module 3—Processes during a face-to-face discussion (with Notes)
1. Presenting the sign invitation for dialogue (Figure 3).
2. As soon as someone has accepted the invitation, open the wallet case and present the iPhone X (the Notes app with the introductory text displayed) to that person.
3. During a discussion, tap Return **before and after** the messages of the communication partner.
4. If the focus of VoiceOver is outside the text box, write; "p" and space. The communication partner with normal vision will then be asked to put the cursor back inside the text box. The person with deafblindness will be able to put the cursor back at the end of the line and press the Return key to continue the conversation.

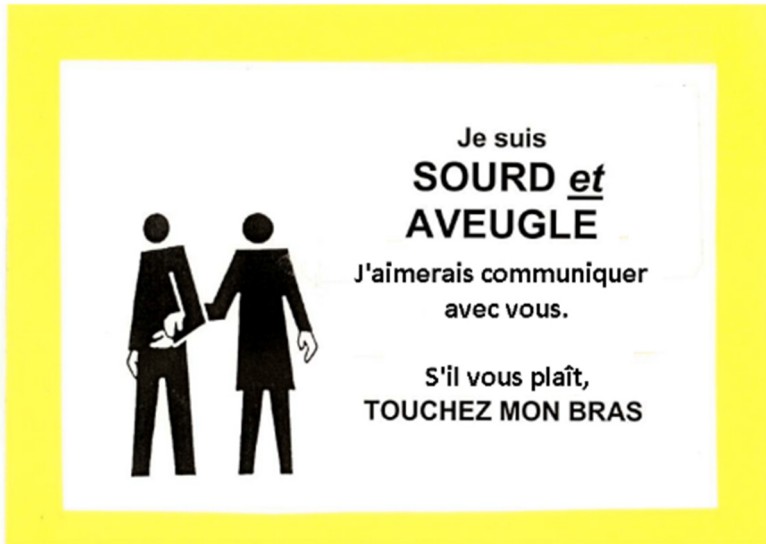

**Figure 3.** New proposition of the French Sign inviting dialogue, from [20] (p. 6).

## 4. Discussion

The purpose of the present study was to explore the feasibility of an innovative communication technology for persons living with deafblindness when wanting to exchange information with a member of the public. We compared our experience to the only study—Cantin et al.—available with a technology test in society (e.g., in a restaurant). The waitress already knew the participant [14] while in our study the attendant at a kiosk, the passer-by, and the security agent were strangers. Here, our results indicate an added value to the scientific literature on the use of tactile technology with "unknown" communication partners. In both studies, the waitress and our communication partners in the shopping mall were happy to help. The participant was deafblind and a super user of technology with a braille display [14] while our participant was a blind clinician simulating deafness using a braille display and an iPhone. Technical problems occurred with the manipulation of the iPhone and HumanWare software in the restaurant [14] while there were no technical problems

with the iPhone, VoiceOver, and Notes, but our participant was not a new user. Here, our results indicate an added value to the scientific literature on the use of tactile technology that is already marketed or commercialized. Our preliminary results propose technical recommendations for training adults over 50 years old with USH1 with communication alternatives. Here, our results are applications of available knowledge for the scientific literature and, for clinicians, to limit device abandonment in deafblindness [8]. Satisfaction of the communication partner with normal vision in the trials also demonstrates that the use of discrete technologies for communication may facilitate social inclusion, instead of other technologies as suggested in a study regarding individuals living with deafblindness, stigma, and the use of communication and mobility assistive devices [21]. The deafblind person in one study [14] confirmed in a subsequent paper [22] that using generally available applications provides access to a normalized life and results in better self-esteem, among other things. Lead users such as this person facilitate future professional interventions in the field by providing positive experiences and successes that make the general public aware of the situation of persons using adapted technologies.

The results of our study (without a deafblind person) show that spelling and punctuation errors, as well as failure to follow the instructions to press the "Return" key after messages, did not affect communication. These results suggest that the face-to-face communication solution we developed is suitable with communication partners with normal vision. However, we must consider that only three tests were carried out and that the communication partner playing the role of the person living with deafblindness was very experienced in the operation of devices and software integrating the communication solution. Nonetheless, her sentences were simple (e.g., "is *Le Garage* shop here?"), and she never sent incomplete, meaningless sentences to the communication partner.

Even with these limitations of the technology, the tests revealed opportunities for improvements of the interface with the communication partner. These improvements concern the sign that invited potential communication partners into dialogue and the organization of textual exchanges.

### 4.1. Dialogue Invitation Sign

Even though all the communication partners understood and reacted well to the sign used in this study in Figure 1, experience showed that it could be more specific to the expected purpose, i.e., to invite a person from the public to a face-to-face conversation. While keeping the current presentation style and icon, we propose a sequence of more specific sentences: *I am DEAF <u>and</u> BLIND/I would like to communicate with you/Please TOUCH MY ARM* (see Figure 3—in French).

### 4.2. Organization of Textual Exchanges

An important opportunity for improvement concerns the organization of messages from both participants (deafblind and communication partner) in the editing area of iOS Notes. As shown in the screenshot in Figure 4a, the rereading of the discussion of Test 1, with the messages of different origins all mixed together, is not easy. The instruction to touch Return at the end of a message aims to introduce a blank line between the texts and ensure a more obvious separation. Its purpose is to facilitate the rereading of discussions, either during or after exchanges.

Strict adherence to the instructions to touch Return after messages would make it easier to replay discussions, both for a person with deafblindness and for a sighted person (see Figure 4b). Without the ability to control the behavior of the communication partner, we propose that during their training, the person with deafblindness needs to be instructed to touch on the return BEFORE and AFTER his messages. Being freed from this task, the initial instructions that would be presented to the communication partner would be simplified.

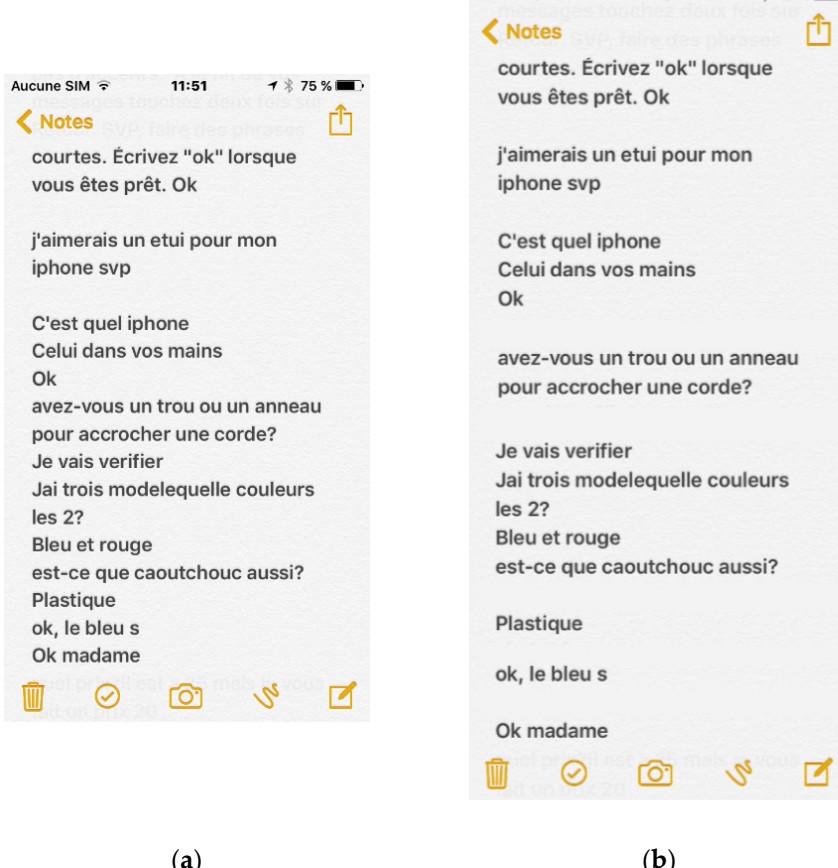

(**a**)　　　　　　　　　　　　(**b**)

**Figure 4.** (**a**) Screenshot of the discussion text messages from Test 1 on iOS Notes. (**b**) Test 1 discussion text messages organized by setting up returns at the end of messages.

> *"Hello, I am deaf and blind. Use this device to communicate with me. Write on the keyboard, but avoid touching for a long time, swiping your finger and do not use accents. Please, make short sentences. Write "ok" when you're ready"*

### 4.3. Strengths and Limitations

The strengths of this study lie in the rigor of its qualitative research methodology, given the use of reflexivity, peer debriefing, and using the participants' direct quotes [23], all available in the French research reports [17–20]. We extracted, translated, and synthetized some data to write this article. In addition, we believe that we demonstrated dependability and trustworthiness, given that the findings appear to be consistent and repeatable [23]. A technology review was carried out and then scrutinized by 15 experts in vision and hearing rehabilitation, research, and engineering to select the best technologies. The entire R&D component utilized an integrated knowledge translation approach [24] by engaging partners in a sensory rehabilitation center, and three reports [18–20] were written in French by the engineer and two clinicians, including the description and justification of the techno-logical choices, operational strategies to ensure the interconnection between the equipment, and the carrying out of communicational exchanges with them. The technological pre-test presented in this article was carried out in an ecological environment, representing a mi-crocosm of the society, in this case, a shopping center near the rehabilitation center. The tests suggest that the face-to-face communication solution we developed is suitable for communication partners with normal vision. The last report [20] was written following the experiment, including all observations as well as recommendations for improvements and for training with USH1. We believe that the figures illustrating communication and

the table with typical communicational exchanges are sufficiently detailed to allow the transferability of data from the use of technologies to another similar social context [23].

The limitations of this study are inherent in the choice of the research designs (R&D and pre-test) and concern external validity. A limited number of tests were conducted and only in one setting. Because no keyboard input incidents had occurred, we were unable to test the recovery procedure for this type of incident. The user playing the role of a person with deafblindness in the tests perfectly mastered the technology integrating the communication solution. She did not commit any faults that could cause a possible breakdown of communication. There is possible bias here about simulated hearing impairment as her hearing was not blocked during the experiment.

### 4.4. Future Research

For the practical dimensions of training (with what personal, financial, and other resources can carry out the recommendations indicated by the clinicians), the second year of our project aims to measure effectiveness, impact on social participation, and cost of a communication technology alternative proposed to two clients newly living with deafblindness. A case study is ongoing with members of our research team in two different rehabilitation centers with different clinicians, involving two adults with USH1 and their caregiver. Data collection starts 3 months before the delivery of the technologies and at 0, 3, and 9 months. Questionnaires and interviews focus on social participation, communication exchanges, experience with training and technologies (facilitators and barriers), costs (personal, technology), and comparison with scenarios with old technologies and without technologies. Future research should document face-to-face communication with more than two adults with USH1 in different societal contexts (e.g., command a meal in a restaurant): how effective and efficient is the technology solution compared to persons without disability? Future research should demonstrate barriers and facilitators for assistive technologies that influence social participation in adults with deafblindness [25].

### 5. Conclusions

Three communication trials in a shopping center demonstrated that it is possible to use a tactile technological solution coupled with a smartphone to interact with unknown communication partners with normal vision, request the location of specific shops, and purchase a product. Due to the involvement of clinicians in research and development with an engineer, technology training modules are recommended to people with USH1 who live with a vision impairment, who have basic knowledge of braille, and who want to communicate face-to-face by means of an iPhone and a braille keyboard. Clinicians proposed a sequence of three training modules in 9 to 12 sessions: prior knowledge (basic operations for iPhone X, VoiceOver, Notes, and Mail, braille display Focus 40) and methods for preparing a face-to-face discussion (with Notes). This training is currently in the validation phase in the second part of an on-going case study over a period of 9 months.

**Author Contributions:** Please note that ( ) are not authors of the paper but they are reported in the acknowledgments, for their direct research participation at different moments. Conceptualization, C.V., B.A., W.W., (N.B.), M.H. and F.B.; Methodology, C.V., B.A., W.W., (N.B.), M.H., F.B. and (S.C.); Software, (W.C.); Validation, (M.C.L.), (S.T.), (W.C.) and (S.C.); Formal Analysis, (W.C.); Investigation, C.V.; Resources, C.V.; Data Curation, (W.C.); Writing—Original Draft Preparation, C.V.; Writing—Review & Editing, C.V., B.A., W.W., M.H. and F.B.; Supervision, C.V. and (M.E.S.); Project Administration, C.V.; Funding Acquisition, C.V. and W.W. All authors have read and agreed to the published version of the manuscript.

**Funding:** This project was funded by the Fondation Envue, Intersectoral Initiative Inclusive Society of the Quebec research fund, the Fonds de recherche en déficience auditive and the Fondation Élan. Wittich is funded by a Junior 2 FRQ-S chercheur boursier career award (# 281454).

**Institutional Review Board Statement:** This project was approved by the research ethics board of the Centre for Interdisciplinary Research in Rehabilitation of Greater Montreal (CRIR-1399-0319 multi).

**Informed Consent Statement:** Informed consent was obtained from all subjects involved in the study.

**Data Availability Statement:** Original data are in French in references [17–20]. They are available from first author at claude.vincent@rea.ulaval.ca.

**Acknowledgments:** The authors wish to warmly thank the following persons for their involvement to the project: Marie-Claire Lemire, Suzanne Trudeau and Walter Cybis, from the Institute Nazareth et Louis-Braille/INLB du Centre intégré de santé et de services sociaux de la Montérégie-Centre and the Institute Raymond Dewar/IRD du Centre intégré universitaire de santé et de services sociaux du Centre-Sud-de-l'Île-de-Montréal. The completion of this project would not have been possible without their important contribution. Thank you to Marie-Eve Schmouth, our research coordinator, for maintaining the communication so well between two cities and among all members of the research team located at the Cirris and the CRIR. Thank you to Normand Boucher, a Cirris researcher that will be involved in the social participation assessment for the next step of the project. Thank you to Sylvie Cantin, our methodologist involved in the next step of the project, for the validation of communication exchanges in society. The authors want to thank all the participants from the focus group for the choice of the technologies, the clinicians as well as public programs for vision and hearing (Josiane Arseneau, Johanne Cantin, Danielle Cloutier, Marieve Cloutier, Jean-Francois Dauteuil, Julie Dufour, Johannie Fex, Bernadette Gavouyère, Christine Levesque, Valérie Roy-Turcotte). Representatives of government pay programs were invited as observers for the focus group; thank you to Benjamin Brisson-Gauthier (hearing aid program) and Nancy Vallée (visual aid program).

**Conflicts of Interest:** The authors declare no conflict of interest.

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
