# Peer review of "Shopping When You Are Deafblind: A Pre-Technology Test of New Methods for Face-to-Face Communication—Deafblindness and Face-to-Face Communication"

_societies, doi:10.3390/soc11040131_

Round 1

Reviewer 1 Report

This is an interesting paper and provide useful information in communication trials  for persons living with deafblindness. Many minor suggestions as following:

  1. the selection bias of the participants should discuss in the content.
  2. the indicator of acceptability of this devise for the participants should evaluate in the final development stage.

Author Response

Response to Reviewer 1 Comments

This is an interesting paper and provide useful information in communication trials for persons living with deafblindness. Many minor suggestions as following:

Point 1: the selection bias of the participants should discuss in the content.

Response 1: Given the type of research design (applied research and experimental development), the same participants that adapted the technology solution in clinic volunteered for the pre-test. This means that “selection bias” is needed here, since it is not a question of having a representative sample but of establishing a proof-of-concept in social context. Since there were no deafblind person in the development phase, a clinician with no vision simulated one. The three partners with normal vision in the shopping were chosen spontaneously on the site, upon their availability.

This text has been added under Participants, p. 8.

Point 2: the indicator of acceptability of this devise for the participants should evaluate in the final development stage

Response 2: As this is a simulation in a shopping mall regarding equipment use by a deafblind person, acceptability cannot be documented as an indicator. We have only documented Level of satisfaction with the exchanges and feelings after exchanges. P. 10. In next phase of the research (case study), there will be indicators of acceptability (See Future research section). No change in text.

Reviewer 2 Report

First of all, I would like to congratulate the authors on this project. This was an interesting manuscript. The article is well structured and presented. I have a few minor recommendations:

  1. The title is a little bit longer, and some keywords are redundant.
  2. In section 2.1 Research design
    1. I recommend inserting a paragraph that explains better experimental development design (Line 87)
    2. It is necessary to make very clear the aim and the objectives of the study. Probably 3.1, 3.2, 3.3 can be the objectives of the study?
    3. In Line 52: „From the 10 studies consulted”… Wich are the 10 studies consulted?

I wish you success with your project.

Author Response

Response to Reviewer 2 Comments

First of all, I would like to congratulate the authors on this project. This was an interesting manuscript. The article is well structured and presented. I have a few minor recommendations:

Point 1: The title is a little bit longer, and some keywords are redundant.

Response 1: There are 13 words in our title (Shopping when you're deafblind:  a pre-technology test of new methods for face-to-face communication).   A good title should include the main variables (shopping, face-to-face communication, technology), the population (deafblind), and the orientation of the research (a pre-technology test of new methods). It seems difficult to cut a word in the title without losing sense.

Key words that will remain are: Deafblindness; Dual sensory loss; Usher syndrome; Assistive communication technology; iPhone; Braille notetaker; Living laboratory. We agree to delete “Assessment; Usability; Effectiveness” because they are only cited in Introduction.

Point 2:  In section 2.1 Research design, I recommend inserting a paragraph that explains better experimental development design (Line 87).

Response 2: We have made a link with the definition and the steps. What is in red is what is reorganised or added.

Applied research and experimental development design (1st design)

“Research and experimental development” design is defined as a “creative and systematic work undertaken in order to increase the stock of knowledge—including knowledge of humankind, culture and society—and to devise new applications of available knowledge” [16]. In the first year of the project, the two creative and systematic steps were Selection of a communication technology and Adaptation. The new applications of available knowledge are presented in the Technology solution section.

Point 3:  In section 2.1 Research design It is necessary to make very clear the aim and the objectives of the study. Probably 3.1, 3.2, 3.3 can be the objectives of the study?

Response 3: The aim is: to explore the feasibility of using a braille display and a smartphone in society, to improve face-to-face communication for a person living with deafblindness, using a simulated communication situation. Research objectives are1) to demonstrate the feasibility of using a braille display and a smartphone in simulated communicational and social exchange; and 2) to propose clinical recommendations for eventually training persons with USH1 for face-to-face communication in society. In the abstract, we have specified the aim (goal) with the verb “aimed”. In the article, we rewrite the same aim (P.3). We have highlighted two research objectives (p.4). In the results section, we added Feasibility and Proposing a training program to make clearer the 2 research objectives.

Point 4:  In section 2.1 Research design In Line 52: „From the 10 studies consulted”… Wich are the 10 studies consulted?

Response 4: From the 10 studies consulted in that recent scoping review [8], usability was challenged in devices that rely on the ‘other’ sense…

I wish you success with your project.

Reviewer 3 Report

The reviewed paper shows preliminary reports from a project testing the usefulness of a way for people with acquired deafblindness to communicate. The topic is interesting and undoubtedly valuable to the broadly understood social functioning of this group of people with disabilities. I believe, however, that the data collected from the conducted “experimental” situation should be treated as an introduction to the main experiment. This main experiment should be carried out in various places where it is possible to test the usefulness of supported communication with deafblind people, appropriately selected according to scientific requirements. The described “experimental” situation included professionals with some experience of working with deafblind people, one of whom was blind, while the other had no visual or hearing problems (a third non-disabled person collected the data). Thus, this experiment does not provide knowledge from the perspective of future users of the system. The data analysis focused on satisfaction with the communication assessed from the perspective of a non-disabled person. In my opinion, a larger-scale experiment should mainly focus on the experiences of deafblind people. Further suggestions include:

  1. Please clarify in the research problems that this is a pre-test of the usability of the communication way assessed by non-disabled users (passersby).
  2. Add information on which clinicians were involved in the project (e.g., what specialty, where they were employed). Why were they the only ones who authored the outline of the program for the deafblind?
  3. My doubts concern the program developed by the clinicians. The study goals refer to recommendations (p.2, also p. 7), the title of the subsection refers to a “training program” (p. 6). There are modules with detailed steps/instructions for action in Table 2.
  4. It is worth specifying who, where, with what personal, financial, and other resources can carry out the recommendations indicated by the clinicians. This is an important part of the whole project and its practical dimension.
  5. Any conclusions should be accompanied by information about the fact that the “experimental” situation did not include deaf-blind people (e.g.“Our results indicated that there were no difficulties occurring with the use of tactile and smartphone technologies in an environment such as a shopping mall”, p. 7; “The results of our study show that spelling and punctuation errors, as well as failure to follow the instructions to press the "Return" key after messages, did not affect communication. These results suggest that the face-to-face communication solution we developed is suitable with communication partners with normal vision.”, p. 8).
  6. Please specify where the reflexivity and the quotes from the respondents can be found in the manuscript, "The strengths of this study lie in the rigor of its qualitative research methodology, given the use of reflexivity, peer debriefing and using the participants' direct quotes in all reports [23]." (p. 8).
  7. In the "Future research" section, it is worth suggesting specific directions for further inquiry into the topic that may be important to other researchers.

Author Response

Response to Reviewer 3 Comments

The reviewed paper shows preliminary reports from a project testing the usefulness of a way for people with acquired deafblindness to communicate. The topic is interesting and undoubtedly valuable to the broadly understood social functioning of this group of people with disabilities. I believe, however, that the data collected from the conducted “experimental” situation should be treated as an introduction to the main experiment. This main experiment should be carried out in various places where it is possible to test the usefulness of supported communication with deafblind people, appropriately selected according to scientific requirements. The described “experimental” situation included professionals with some experience of working with deafblind people, one of whom was blind, while the other had no visual or hearing problems (a third non-disabled person collected the data). Thus, this experiment does not provide knowledge from the perspective of future users of the system. The data analysis focused on satisfaction with the communication assessed from the perspective of a non-disabled person. In my opinion, a larger-scale experiment should mainly focus on the experiences of deafblind people. Further suggestions include:

Point 1:  Please clarify in the research problems that this is a pre-test of the usability of the communication way assessed by non-disabled users (passersby).

Response 1: “The setting described in this study is a pre-requisite to further tests of usability by deafblind people.” Before measuring usability, we must test the technology in simulating face to face communication in a shopping mall, in real context, with unfamiliar person. To clarify, we have deleted key words (usability, effectiveness and evaluation). See response 3 as well.

Point 2: Add information on which clinicians were involved in the project (e.g., what specialty, where they were employed). Why were they the only ones who authored the outline of the program for the deafblind?

Response 2: In Participants section, we have completed (see underlined). Location of their work be available in the Acknowledgements section (under confidentiality).

There were three R&D participants, employed by two rehab centers located in the Montreal area. The low vision therapist, who is also blind, simulated the person living with deafblindness in the shopping mall. The special educator, competent in sign language, participated in the test planning and on-site analysis and discussions in the mall. The engineer took notes on the events occurring during and after the tests. The community participants were an attendant at a kiosk, a passer-by and a security agent.  They were three communication partners (interlocutors) with normal vision that were interviewed after the tests in the shopping mall. 

Point 3: My doubts concern the program developed by the clinicians. The study goals refer to recommendations (p.2, also p. 7), the title of the subsection refers to a “training program” (p. 6). There are modules with detailed steps/instructions for action in Table 2.

Response 3: In fact, that was not clear, we have completed (see underlined).  We have uniformized the aims (abstract and article). The research objectives in the article are: 1) to demonstrate the feasibility of using a braille display and a smartphone in simulated communicational and social exchange; and 2) to propose clinical recommendations for eventually training persons with USH1 for face-to-face communication in society. In results, we present result that reinforce Feasibility and Clinical recommendations for eventually training. In Table 2, we have modified the title: Sequence of three training modules proposed to new clients with deafblindness.

Point 4:  It is worth specifying who, where, with what personal, financial, and other resources can carry out the recommendations indicated by the clinicians. This is an important part of the whole project and its practical dimension.

Response 4: In Future research, we have we have completed (see underlined) : For practical dimensions of training (with what personal, financial, and other resources can carry out the recommendations indicated by the clinicians), the second year of our project aims to measure effectiveness, impact on social participation and cost of a communication technology alternative proposed to two newly clients living with deafblindness. A case study is ongoing with members of our research team, in two different rehabilitation centers with different clinicians, involving two adults with USH1 and their caregiver. Data collection starts 3 months before the delivery of the technologies, and at 0, 3 and 9 months. Questionnaires and interviews focus on social participation, communication exchanges, experience with training and technologies (facilitators and barriers), costs (personals, technology) and comparison with scenarios with old technologies and without technologies will be filled.

Point 5: Any conclusions should be accompanied by information about the fact that the “experimental” situation did not include deaf-blind people (e.g.“Our results indicated that there were no difficulties occurring with the use of tactile and smartphone technologies in an environment such as a shopping mall”, p. 7; “The results of our study show that spelling and punctuation errors, as well as failure to follow the instructions to press the "Return" key after messages, did not affect communication. These results suggest that the face-to-face communication solution we developed is suitable with communication partners with normal vision.”, p. 8).

Response 5: In Discussion, p.11, all paragraph was modified in consequence (see Reviewer 4, point 1)

Point 6:  Please specify where the reflexivity and the quotes from the respondents can be found in the manuscript, "The strengths of this study lie in the rigor of its qualitative research methodology, given the use of reflexivity, peer debriefing and using the participants' direct quotes in all reports [23]." (p. 8).

Response 6: We have completed (see underlined). The strengths of this study lie in the rigor of its qualitative research methodology, given the use of reflexivity, peer debriefing and using the participants' direct quotes [23], all available in the French research reports [17, 18, 19 and 20]. We have extracted, translated, and synthetize some data to write this article.

Point 7:  In the "Future research" section, it is worth suggesting specific directions for further inquiry into the topic that may be important to other researchers.

Response 7: We have completed (see underlined). Future research should document face to face communication with more than 2 adults with USH1, in different societal contexts (e.g., command a meal in a restaurant): how effective and efficient is the technology solution compared to persons without disability? Future research should demonstrate barriers and facilitators for assistive technologies that influence social participation in adults with deafblindness [25].

Reviewer 4 Report

This paper presented a solution to help deaf-blind people using a braille display and a smartphone in simulated communicational and social exchange. The communication trials in a shopping center showed the importance of using the tactile technological solution coupled with a smartphone.  It helped them interact with partners with normal vision and request the location of specific shops to purchase a product.

Overall:

  1. The article presented a novel idea to help people with deaf-blind using new technology.
  2. The paper is organized well with proper structure, and the bibliography is sufficient and well given.
  3. The presented methodology and the results are communicated. The novel contribution of the paper is highlighted.
  4. The experimental design is reasonable, and the evaluation results show the superiority of the proposed scheme.

This paper can be accepted. However, there are some points that the authors can handle:

  1. You mentioned in the discussion that you compared your work with reference number [15]. You can give more details about that, how your work is better than it.
  2. It would be better for future work to test your work with real deafblind people not simulated ones because they can act in a different way.
  3. It would be better for future work to add more experiments as three are not enough.
  4. On page 5, you mentioned looking at figure 3-a but I didn’t find it. Maybe you missed adding it to the article.
  5. The following papers are recently done and related to your work so, the authors can add them as reference.
    1. A Novel Marker Detection System for People with Visual Impairment Using the Improved Tiny-YOLOv3 Model
    2. An Outdoor Navigation System for Blind Pedestrians Using GPS and Tactile-Foot Feedback.
    3. Making shopping easy for people with visual impairment using mobile assistive technologies.
    4. Ebsar: Indoor guidance for the visually impaired.

Author Response

Response to Reviewer 4 Comments

This paper presented a solution to help deaf-blind people using a braille display and a smartphone in simulated communicational and social exchange. The communication trials in a shopping center showed the importance of using the tactile technological solution coupled with a smartphone.  It helped them interact with partners with normal vision and request the location of specific shops to purchase a product.

Overall:

  1. The article presented a novel idea to help people with deaf-blind using new technology.
  2. The paper is organized well with proper structure, and the bibliography is sufficient and well given.
  3. The presented methodology and the results are communicated. The novel contribution of the paper is highlighted.
  4. The experimental design is reasonable, and the evaluation results show the superiority of the proposed scheme.

This paper can be accepted. However, there are some points that the authors can handle:

Point 1:  You mentioned in the discussion that you compared your work with reference number [15]. You can give more details about that, how your work is better than it.

Response 1: I think you mean reference number [14].  The paragraph was rewritten. We have compared our experience to the only study ─Cantin et al─ available with a technology test in society (e.g., in a restaurant). The waitress already knew the participant [14] while in our study the attendant at a kiosk, the passer-by and the security agent were strangers. Here, our results indicate an added value to the scientific literature on the use of tactile technology with “unknown” communication partners. In both studies, the waitress and our communication partners in the shopping mall were happy to help. The participant was deafblind and a super user of technology with a braille display [14] while our participant was a blind clinician simulating a deafness using a braille display and an iPhone. Technical problems occurred with the manipulation of the iPhone and HumanWare software in the restaurant [14] while there were no technical problems with the iPhone, VoiceOver and Notes, but the participant was not a new user. Here, our results indicate an added value to the scientific literature on the use of tactile technology that is already marketed or commercialized. Our preliminary results proposed technical recommendations for training adults over 50 years old with USH1 with communication alternatives. Here, our results are applications of available knowledge for the scientific literature and, for clinicians, to limit device abandonment in deafblindness [8].

Point 2:  It would be better for future work to test your work with real deafblind people not simulated ones because they can act in a different way.

Response 2: It was already indicated, in Future research. We have upgraded the first paragraph (see underlined). For practical dimensions of training (with what personal, financial, and other resources can carry out the recommendations indicated by the clinicians), the second year of our project aims to measure effectiveness, impact on social participation and cost of a communication technology alternative proposed to two newly clients living with deafblindness.

Also, we have added this: Future research should document face to face communication with more than two adults with USH1, in different societal contexts (e.g., command a meal in a restaurant): how effective and efficient is the technology solution compared to persons without disability?

Point 3:  It would be better for future work to add more experiments as three are not enough.

Response 3: See response 2.

Point 4:  On page 5, you mentioned looking at figure 3-a but I didn’t find it. Maybe you missed adding it to the article.

Response 4. It appears under the section Organization of textual exchanges.

Point 5:  The following papers are recently done and related to your work so, the authors can add them as reference.

    1. A Novel Marker Detection System for People with Visual Impairment Using the Improved Tiny-YOLOv3 Model
    2. An Outdoor Navigation System for Blind Pedestrians Using GPS and Tactile-Foot Feedback.
    3. Making shopping easy for people with visual impairment using mobile assistive technologies.
    4. Ebsar: Indoor guidance for the visually impaired.

Response 5: All of those technologies are developed for blind persons, and for their mobility (not for face-to-face communication).  Those technology can “talk” but they are not tactile or vibratory for communication. Most of the time, persons with only visual problem can use those technologies in wearing ear plugs, by listening verbal or robotic indications or someone at distance is reading the environment with a camera. But persons with USH1, they can’t hear from their birth, and they only learn braille when they lose their sight, around 50 years old. So, they can’t use technologies developed only for visual impairment, because they can’t ear the voice coming from them.  

See p. 3 where it is mentioned that technology is often conceived only for vision impairment or hearing impairment but not for dual sensory. Haptic and tactile aids are needed. No text has been added.
